# Theoretical Study of Hydrogen Production from Ammonia Borane Catalyzed by Metal and Non-Metal Diatom-Doped Cobalt Phosphide

**DOI:** 10.3390/molecules27238206

**Published:** 2022-11-24

**Authors:** Dong-Heng Li, Qiao-Mei Li, Shuang-Ling Qi, Hai-Chuan Qin, Xiao-Qin Liang, Laicai Li

**Affiliations:** College of Chemistry and Material Science, Sichuan Normal University, Chengdu 610068, China

**Keywords:** ammonia borane, hydrogen evolution reaction, CoP, diatom-doped, density functional theory

## Abstract

The decomposition of ammonia borane (NH_3_BH_3_) to produce hydrogen has developed a promising technology to alleviate the energy crisis. In this paper, metal and non-metal diatom-doped CoP as catalyst was applied to study hydrogen evolution from NH_3_BH_3_ by density functional theory (DFT) calculations. Herein, five catalysts were investigated in detail: pristine CoP, Ni- and N-doped CoP (CoP_Ni-N_), Ga- and N-doped CoP (CoP_Ga-N_), Ni- and S-doped CoP (CoP_Ni-S_), and Zn- and S-doped CoP (CoP_Zn-S_). Firstly, the stable adsorption structure and adsorption energy of NH_3_BH_3_ on each catalytic slab were obtained. Additionally, the charge density differences (CDD) between NH_3_BH_3_ and the five different catalysts were calculated, which revealed the interaction between the NH_3_BH_3_ and the catalytic slab. Then, four different reaction pathways were designed for the five catalysts to discuss the catalytic mechanism of hydrogen evolution. By calculating the activation energies of the control steps of the four reaction pathways, the optimal reaction pathways of each catalyst were found. For the five catalysts, the optimal reaction pathways and activation energies are different from each other. Compared with undoped CoP, it can be seen that CoP_Ga-N_, CoP_Ni-S_, and CoP_Zn-S_ can better contribute hydrogen evolution from NH_3_BH_3_. Finally, the band structures and density of states of the five catalysts were obtained, which manifests that CoP_Ga-N_, CoP_Ni-S_, and CoP_Zn-S_ have high-achieving catalytic activity and further verifies our conclusions. These results can provide theoretical references for the future study of highly active CoP catalytic materials.

## 1. Introduction

The energy crisis and environmental pollution are two major problems humans are faced with in society today. To deal with these severe problems, taking advantage of renewable energy to replace fossil energy is an important strategy for the international community.

As a renewable energy source, hydrogen energy is light in weight, high in heat, non-toxic, harmless, excellent in thermal conductivity, clean, and pollution-free, which makes it a representative of green energy [1,2,3,4]. To date, there are many common industrial hydrogen production methods [5,6,7,8], such as the methods of NH_3_BH_3_ decomposition, water cracking, water and ethanol mixture pulse discharge, etc. However, how to release hydrogen safely and efficiently remains the main obstacle to the spread of hydrogen energy. NH_3_BH_3_ is considered as one of the most ideal hydrogen storage materials because of its non-toxic, easy storage and transportation and reversible dehydrogenation reaction [9,10,11,12]. Catalytic decomposition of NH_3_BH_3_ is accepted as a promising method for hydrogen production. Although traditional catalysts such as platinum-based or rhodium-based noble metal catalysts have high activity for NH_3_BH_3_ decomposition, their application is limited due to high cost and short time [13,14,15]. Therefore, there is an upsurge in research to look for non-noble metal catalysts with high catalytic activity and service life [16,17,18,19]. At present, quantities of composite materials with high catalytic activity and high stability which can catalyze the evolution of hydrogen from ammonia borane have been synthesized [20,21,22,23,24].

Cobalt phosphide (CoP) has become a typical representative of inexpensive transition metal phosphides due to its advantages of low production cost, good stability, and catalytic activity [25,26,27]. CoP materials include CoP nanowire arrays [28,29,30], non-metallic-doped CoP materials [31,32], metal-doped CoP materials, etc. [33,34]. Various doped catalytic materials based on cobalt phosphide have been widely studied as excellent catalysts for NH_3_BH_3_ decomposition to produce hydrogen [35,36,37], such as non-noble metal cobalt phosphide nanometer materials supported by layered porous carbon (CoPNPs), which were synthesized by step-by-step calcination and phosphating, using a cobalt-based organic framework (CO-MOF-74) as template [38], carbon-point-constrained CoP-CoO nanostructured materials with strong interfacial synergies, which trigger the strong hydrogen evolution performance of NH_3_BH_3_ [39], nickel-loaded cobalt phosphide (Ni@CoP) materials, etc. [40,41]. These CoP-based modified catalysts improve the hydrogen evolution performance of NH_3_BH_3_. It was found that different types of doping have an important effect on the catalytic activity of the materials [42,43]. For example, silver and nitrogen diatom-doped zinc oxide has become a salient way to obtain high quality P-type zinc oxide [44]. The dehydrogenation of NH_3_BH_3_ catalyzed by Co and Cu diatom-doped magnesium oxide is better than that of Co- and Cu-doped magnesium oxide alone [45]. The catalytic performance and efficiency of O and Mo diatom-doped cobalt phosphide lamellar nanomaterials as catalysts for water cracking has been significantly improved [46].

In view of CoP as an excellent catalyst for the dehydrogenation of NH_3_BH_3_ and the significant improvement of catalyst performance by diatom-doped metal and non-metal co-doped CoP as a catalyst for hydrogen production from NH_3_BH_3_ was studied in this paper, and the mechanism of its catalytic dehydrogenation is discussed. At the same time, the catalytic activity of different doped catalysts was studied. The study of the physical properties of doped CoP variants is expected to explain the correlation between the physical properties of the catalysts and the catalytic activity of NH_3_BH_3_ dehydrogenation, which provides some theoretical references for the optimization and design of the catalysts for hydrogen production from NH_3_BH_3_.

## 2. Calculation Methods

In this study, all the structure optimization, band structures, and density of states (DOS) were calculated using Dmol3 in the Material Studio 8.0 program developed by Accelrys, Inc. The generalized gradient approximation Perdew–Burke–Ernzerhof (PBE) exchange-correlation functional was adopted and the nuclear electron was described by effective core potential (ECP) [47]. To expand the electronic wave function, the double numerical plus polarization (DNP) basis set was used [48]. A 2 × 2 × 1 k-points was sampled using the Monkhorst–Pack method. On this basis, the energy convergence criterion of the self-consistent iterative process was set to 2 × 10 ^−5^ Ha, the force convergence accuracy was set to 0.004 Ha/Å, and the maximum displacement was set to 0.005 Å. Meanwhile, The LST/QST method was used to search for reaction transition states whose structures were further confirmed by frequency analysis [49].

A 2 × 3 CoP (101) slab model with six-layer-atom was built to represent CoP catalysts, which was consistent with that of Deniel et al. [50] and Cao et al. [51,52]. This model contained 72 Co and 72 P atoms. In order to prevent the interactions between periodic images of the slabs, a 15 Å vacuum layer is added in Z direction. In doped CoP, one of the P atoms on pristine CoP (101) surface were replaced by non-metallic N or S atom, and one of the Co atoms on the pristine CoP (101) surface was replaced by Ni, Ga, or Zn atoms, respectively, as shown in Figure 1. Then, metal and non-metal diatom-doped CoP catalyst models were obtained. Herein, we focus on four different types of doped CoP: Ni and N diatom-doped CoP, as denoted CoP_Ni-N_; Ga and N diatom-doped CoP, denoted CoP_Ga-N_; Ni and S diatom-doped CoP, denoted CoP_Ni-S_; and Zn and S diatom-doped CoP, denoted CoP_Zn-S_.

## 3. Results and Discussions

### 3.1. Adsorption of NH_3_BH_3_ on the Surface of CoP and Its Doped Catalysts

The stable adsorption structure of NH_3_BH_3_ on CoP(101) slabs was obtained by optimizing the model of CoP, as shown in Figure 2. In the optimized adsorption configuration, the H(1) atom on the NH_3_BH_3_ was adsorbed on the Co(3) atom on the CoP(101) surface, and the distance between the H(1) atom and the Co(3) atom was shortened from 1.79 Å to 1.65 Å. However, the bond length of B-H(1) in NH_3_BH_3_ increased from 1.26 Å to 1.28 Å. The electron density map of the adsorption configuration of NH_3_BH_3_ on CoP(101) surface is also shown in Figure 2. It can be seen from Figure 2 that the overlap of electron cloud occurs between H(1) atom of NH_3_BH_3_ and Co(3) atom of CoP(101) surface, indicating that electron interaction occurs between the H(1) atom of NH_3_BH_3_ and Co(3) atom on the CoP(101) surface. In the process of adsorption, a part of the energy is released due to the reduction of molecular motion velocity, and this part of the energy is called adsorption energy (Eads), which can be calculated as:Eads = −(E_total_ − E_slab_ − E_AB_)(1)

In this formula, E_total_, E_slab_, and E_AB_ are potential energies of AB adsorbed on the slab model, the slab model, and AB molecule. The adsorption process of NH_3_BH_3_ on CoP (101) surface is an activation process, and its adsorption energy is −1.19 eV. The other stable adsorption structures of NH_3_BH_3_ absorbed on the surface of four different kinds of diatom-doped CoP catalysts (CoP_Ni-N_, CoP_Ga-N_, CoP_Ni-S,_ and CoP_Zn-S_) are shown in Appendix A. The electron density maps of the adsorption structure are also shown in Appendix A. The adsorption energies of NH_3_BH_3_ on the four types of doped catalysts are −1.22 (CoP_Ni-N_), −1.29 (CoP_Ga-N_), −1.21 (CoP_Ni-S_), and −1.25 eV (CoP_Zn-S_), which indicates that NH_3_BH_3_ can be stably adsorbed on the surface of the four kinds of doped catalysts, and the adsorption process of NH_3_BH_3_ on the surface of the four kinds of diatom-doped catalysts is also an activation process.

### 3.2. Hydrogen Evolution Mechanism of NH_3_BH_3_ on the Surface of the Catalyst

According to our investigation, there are four potential pathways available for the hydrogen evolution reaction of NH_3_BH_3_ on the catalyst surfaces, as shown in Figure 3.

Firstly, NH_3_BH_3_ is adsorbed on the surface of the catalyst to form a stable adsorption reactant denoted as M1. Then, NH_3_BH_3_ is dehydrogenated through four different reaction pathways to obtain the product P1. In the reaction pathway I, the reactant M1 generates the intermediate M2 via the transition state TS1, and then M2 through the transition state TS2 generates the product P1. In this process, one of the B-H bonds of the NH_3_BH_3_ in M1 is broken to become the intermediate M2 via the transition state TS1. Afterwards, another H atom leaves the N atom of the NH_3_BH_2_ intermediate, forming bond with the previously generated H atom, generating the final product H_2_ and finally completing the hydrogen evolution reaction. In reaction pathway II, the first step of reactant M1 to intermediate M2 is the same as pathway I. After that, one of the N-H bonds of the NH_3_BH_3_ in M2 is broken to form the intermediate M3 via the transition state TS3. The configuration of intermediate M3 is that two H atoms are independently adsorbed on the surface of the catalyst. Finally, the two H atoms adsorbed on the surface of the catalyst are combined together to form the final product P1 via transition state TS4. In the reaction pathway III, the first step is the reactant M1 generates the intermediate M3 via the transition state TS5, in which, different from reaction pathway II, the N-H and B-H of NH_3_BH_3_ are broken simultaneously. The following step is that M3 generates the final product P1 via transition state TS4, which is the same as path II. In reaction pathway IV, the M1 directly generates the H_2_ via transition state TS6.

The structural changes involved in the process of NH_3_BH_3_ dehydrogenation catalyzed by the five catalysts are shown in Figure 4 (CoP_Zn-S_), Appendix A (CoP_Ni-S_). Since each reaction pathway is similar for different catalysts, herein only CoP_Zn-S_ is discussed. The relevant structural parameters of the hydrogen evolution reaction catalyzed by CoP_Zn-S_ are listed in Table 1, and the relative energies and activation energies in each step are listed in Table 2 and as shown in Figure 4.

For reaction path I, NH_3_BH_3_ is adsorbed on the catalyst surface to form M1_Zn-S_, and the adsorption site of NH_3_BH_3_ is above the Co(3) atom on the CoP_Zn-S_ surface. Then in M1_Zn-S_, the B-H(1) bond is broken, and the H(1) atom escapes from NH_3_BH_3_ and migrates between the Co(3) and Zn atoms on the surface of CoP_Zn-S_, forming intermediate M2_Zn-S_ via transition state TS1_Zn-S_ with activation energy of 22.71 kcal/mol. In this process, the distance between the B atom and H(1) atom increases from 1.28 Å to 1.81 Å and finally to 2.50 Å from M1_Zn-S_ to TS1_Zn-S_, forming M2_Zn-S_. The distance between H(1) and Co(3) is shortened from 1.65 Å in M1_Zn-S_ to 1.47 Å in TS1_Zn-S_ and then to 1.58 Å in M2_Zn-S_. In the intermediate M2_Zn-S_, the H(1) atom on the surface of the catalyst and the H(2) atom on the N atom of NH_3_BH_3_ converge to form H_2_ via transition state TS2_Zn-S_. The activation energy of TS2_Zn-S_ is 39.56 kcal/mol. The distance between the H(1) atom and Co(3) changes from 1.58 Å to 1.52 Å, and the distance between the H(1) atom and H(2) decreases from 2.41 Å to 1.72 Å,. In the process of M2_Zn-S_→TS2_Zn-S_→P1_Zn-S_, the distance between H(2) and N atoms changes from 1.05 Å to 1.92 Å and finally to 3.66 Å.

For reaction pathway II, firstly, the reactant NH_3_BH_3_ is adsorbed on the surface of the catalyst to form M1_Zn-S_, and the M1_Zn-S,_ through the transition state TS1_Zn-S,_ forms the intermediate M2_Zn-S_, which is the same as the process of M1_Zn-S_→M2_Zn-S_ in pathway I. Subsequently, in the intermediate M2_Zn-S_, the H(2) atom on the N atom of NH_3_BH_3_ is gradually detached from the NH_3_BH_3_ and adsorbed between the Co(3) and Zn on the CoP(101) surface of the catalyst. With the movement of the H(2) atom, the H(1) atom is adsorbed to the upper left of the Co(3) atom to form the intermediate M3_Zn-S_, which is the transition state TS3_Zn-S_ with an activation energy of 42.04 kcal/mol. In this process, the distance between the H(2) atom and the N atom of NH_3_BH_3_ increases from 1.05 Å to 1.51 Å and finally to 2.55 Å in M3_Zn-S_. The distance between the H(1) and Co(3) atoms varies from 1.58 Å for M2_Zn-S_ to 1.54 Å for TS3_Zn-S_ and finally to 1.65 Å for M3_Zn-S_. The distances between H(2) and Co(3), H(2), and Zn vary from 3.31 Å and 2.96 Å in M2_Zn-S_ to 3.15 Å and 2.52 Å in TS3_Zn-S_ and finally to 1.59 Å and 3.05 Å in M3_Zn-S_. In the intermediate M3_Zn-S_, the two H atoms adsorbed on the surface of the catalyst are close to each other via the transition state TS4_Zn-S_ to form product P1. The activation energy of the transition state TS4_Zn-S_ is 9.73 kcal/mol.

For reaction path III, the reactant NH_3_BH_3_ is first adsorbed on the catalyst surface to form M1_Zn-S_. Then, the H(1) and H(2) atoms, respectively cleaved from the B and N atoms of NH_3_BH_3_, were adsorbed on the surface of the catalyst above the Zn atom of Co(3) atom to form the intermediate M3_Zn-S_ through the transition state TS5_Zn-S_ with an activation energy of 28.80 kcal/mol. In this process, the distance between the B atom and H(1) atom increases from 1.28 Å to 0.178 Å, and the distance between the H(2) atom and N atom increases from 1.03 Å to 1.50 Å. The intermediate M3_Zn-S_ through the transition state TS4_Zn-S_ forms the product P1_Zn-S_, which is consistent with the process of M3_Zn-S_→P1_Zn-S_ in path II.

For reaction path IV, in M1_Zn-S_, the H atom on the B atom of NH_3_BH_3_ and the H atom on the N atom of NH_3_BH_3_ directly generated the H_2_ via the transition state TS6_Zn-S_ with the activation energy of 22.15 kcal/mol, in which the distance between the H(1) and H(2) atoms decreases from 2.53 Å to 1.99 Å in TS6_Zn-S_.

The reaction mechanisms of pristine CoP, CoP_Ni-N_, CoP_Ga-N_, or CoP_Ni-S_ catalyzed NH_3_BH_3_ are similar to the CoP_Zn-S_ catalyst. The details of the configuration changes and configuration parameters of the reaction process are shown in Appendix A (CoP_Ni-S_). The changes in structural parameters are shown in Appendix A (CoP_Ni-S_). The results from the configuration changes in the reaction process of Appendix A, can also indicate that the reaction mechanism of CoP and the other diatom-doped CoP catalysts has small differences.

The conclusion can be draw from Table 2 that in the reaction of NH_3_BH_3_ dehydrogenation catalyzed by CoP_Zn-S_, the control steps of each reaction pathway are different, (CoP, CoP_Ni-N_, CoP_Ga-N_, CoP_Ni-S_, as shown in Appendix A) which are M2_Zn-S_→TS2_Zn-S_ (pathway I), M2_Zn-S_→TS3_Zn-S_ (pathway II), M1_Zn-S_→TS5_Zn-S_ (pathway III), and M1_Zn-S_→TS6_Zn-S_ (pathway IV), respectively. The energy barrier values of each control step are 39.56 kcal/mol (pathway I), 42.04 kcal/mol (pathway II), 28.08 (pathway III) kcal/mol and 22.15 kcal/mol (pathway IV), respectively. According to the comparison of activation energy of each reaction path control step, the optimal pathway of the NH_3_BH_3_ dehydrogenation reaction is reaction pathway IV, and the energy barrier of the control step is 22.15 kcal/mol.

Considering the energy changes in the five catalysts, the activation energies of CoP, CoP_Ni-N_, CoP_Ga-N_, CoP_Ni-S,_ and CoP_Zn-S_ catalyzing the decomposition of NH_3_BH_3_ to hydrogen at each step are listed in Table 3. In CoP-catalyzed NH_3_BH_3_ dehydrogenation, as shown in Appendix A, the activation energy of the optimal control step of the four reaction pathways is 31.35 kcal/mol. The activation energies of CoP_Ni-N_, CoP_Ga-N,_ and CoP_Ni-S_ are 27.11 kcal/mol, 23.18 kcal/mol, and 20.67 kcal/mol, respectively. The energy level changes of the five catalysts in the reaction process are shown in Appendix A (CoP_Zn-S_). By comparing the catalytic activities of CoP, CoP_Ni-N_, CoP_Ga-N_, CoP_Ni-S,_ and CoP_Zn-S_, it is found that the simultaneous doping of metal and non-metal with CoP is beneficial to the improvement of NH_3_BH_3_ hydrogen evolution activity. A large number of studies on cobalt-phosphide-modified materials can prove that the dopant of N, S, Ni, Zn, and Ga can improve the catalytic performance of CoP, which is consistent with our theoretical calculation results. For instance, Chen et al. [53] reported that Ni-doped CoP could accelerate the process of hydrogen evolution both in acid and alkaline media, showing excellent electrochemical stability and durability. Li et al. [54] found N and Mo co-doped heteroatoms can optimize the morphology and surface structure of CoP. Anjum et al. [55] studied sulfur-doped cobalt phosphide electrocatalysts and concluded that their performance is better than all-noble-metal electrocatalysts in alkaline electrolyzers for overall water splitting. Yang et al. [56] synthesized Zn-doped CoP nanowire arrays for boosting hydrogen generation, and they found the overpotential of Zn-doped CoP was two times lower than undoped CoP. Zhang et al. [57] also reported that Ga dopant could enhance the activity of CoP.

### 3.3. Performance Calculation of Catalysts

The stable catalyst models of CoP, CoP_Ni-N_, CoP_Ga-N_, CoP_Ni-S,_ and CoP_Zn-S_ were optimized and obtained. The band structure and density of states (DOS) of the stable catalyst were calculated, as shown in Appendix A. The diagram of band structure is marked by 1 on the left, and the map of density of states is marked by 2 on the right, in which a1, b1, c1, d, and e1 represent the band structure diagram of CoP, CoP_Ni-N_, CoP_Ga-N_, CoP_Ni-S,_ and CoP_Zn-S,_ respectively. Meanwhile, a2, b2, c2, d2, and e2 represent the map of density of states of CoP, CoP_Ni-N_, CoP_Ga-N_, CoP_Ni-S,_ and CoP_Zn-S_. The red dashed line in the figure represents the Fermi level.

The DOS map, which refers to the number of states in a unit frequency interval, is used to characterize the distribution of electron cloud density near the Fermi level. The Fermi level is a parameter used to measure the catalytic activity, and its value is the average of the sum of the highest energy occupied orbital and the lowest energy occupied orbital. The greater the density of the electron cloud near the Fermi level, the stronger the catalytic activity of the material. We calculated the total DOS of the five catalysts at the Fermi level as follows: 37.4 (CoP), 41.9 (CoP_Ni-N_), 42.8 (CoP_Ga-N_), 43.8 (CoP_Ni-S_), and 42.1 (CoP_Zn-S_), indicating that the electron cloud density near the Fermi level of CoP_Ni-N_, CoP_Ga-N_, CoP_Ni-S,_ and CoP_Zn-S_ increases compared with pristine CoP. This may be the reason why CoP_Ni-N,_ CoP_Ga-N_, CoP_Ni-S,_ and CoP_Zn-S_ are able to improve the activity of NH_3_BH_3_ hydrogen evolution. Sun et al. reported similar studies on polysulfur confinement and the electrochemical kinetics of amorphous cobalt phosphide-enhanced lithium-sulfur batteries [58].

In the end, to research the effect of interfacial adsorption on catalytic activity for the five doped catalysts, the charge density difference (CDD) between NH_3_BH_3_ and the five different catalysts were calculated. CDD is one of the important methods to study electronic structure. The electron flow direction after the interaction of each segment can be intuitively obtained, or the change of electron density during the formation of atoms into molecules, and the nature of chemical bonds can be explored. As shown in Figure 5, blue is the electron accumulation, while red represents the electron depletion. From Figure 5, we can realize the charge transfer characteristics of NH_3_BH_3_ and the five catalytic adsorption processes. Compared with undoped CoP, the electronic interaction between doped catalysts and NH_3_BH_3_ is enhanced, which indicates the strong electronic interaction between the catalyst and NH_3_BH_3,_ determining the catalytic activity in the adsorption process.

## 4. Conclusions

In this paper, metal and non-metal diatom-doped CoP as catalyst was applied to study hydrogen evolution from NH_3_BH_3_ by DFT calculations. The doped catalysts involved in CoP_Ni-N_, CoP_Ga-N_, CoP_Ni-S,_ and CoP_Zn-S_ were formed by replacing Co atoms with Ni, Ga, or Zn, and P atoms with S or N on the surface of the CoP(101), respectively. First of all, the adsorption process of NH_3_BH_3_ on each catalyst was explored, and the adsorption energy and electron density maps were obtained. From the values of adsorption energies and electron density maps, the conclusion can be drawn that each doped type of catalyst has a strong adsorption effect on NH_3_BH_3_, which is activated on the surface of the catalyst. Then, we further studied the reaction mechanism of the decomposition of NH_3_BH_3_ into H_2_ and NH_2_BH_2_ catalyzed by five catalysts (CoP, CoP_Ni-N_, CoP_Ga-N_, CoP_Ni-S,_ and CoP_Zn-S_). In this investigation, four pathways were designed, and the best reaction pathways for each catalyst were found. By analyzing the activation energy of the control step, it can be seen clearly that the energy barrier values of the control step for the five catalysts are Ea (CoP) > Ea (CoP_Ni-N_) > Ea (CoP_Ga-N_) > Ea (CoP_Zn-S_) > Ea (CoP_Ni-S_). According to the energy barrier results, the activity of the five catalysts should be CoP_Ni-S_ > CoP_Zn-S_ > CoP_Ga-N_ > CoP_Ni-N_. Finally, the structural performance of the catalyst was investigated, and the band structure and DOS of the CoP_Ni-N_, CoP_Ga-N_, CoP_Ni-S_, and CoP_Zn-S_ catalysts were calculated. The total DOS of the five catalysts at the Fermi level are 37.4 (CoP), 41.1 (CoP_Ni-N_), 42.8 (CoP_Ga-N_), 43.8 (CoP_Ni-S_), and 42.1 (CoP_Zn-S_). The study results we obtained have revealed the relationship between the physical properties of doped CoP materials and their catalytic activities, which provides theoretical support for a large number of high-activity cobalt phosphide materials doped with non-metals (N, S) and metals (Ni, Ga, Zn) and references for the future study of highly active CoP catalytic materials.

## Figures and Tables

**Figure 1 molecules-27-08206-f001:**
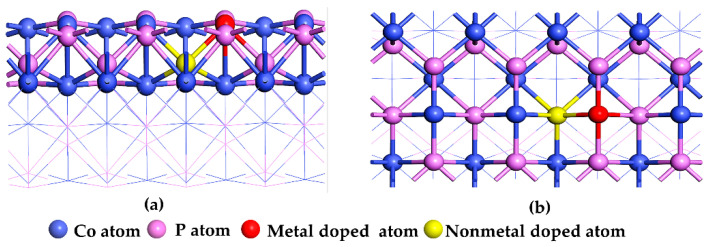
Front view (**a**) and top view (**b**) of metal- and non-metal-doped CoP catalyst.

**Figure 2 molecules-27-08206-f002:**
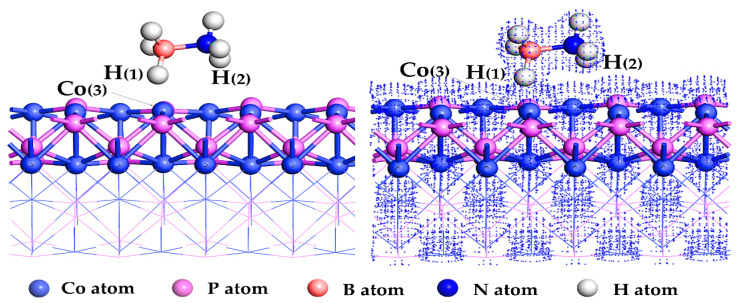
The left shows its adsorption configuration and the right shows the electron density diagram of the adsorption configuration of ammonia borane on the CoP(101) plane.

**Figure 3 molecules-27-08206-f003:**
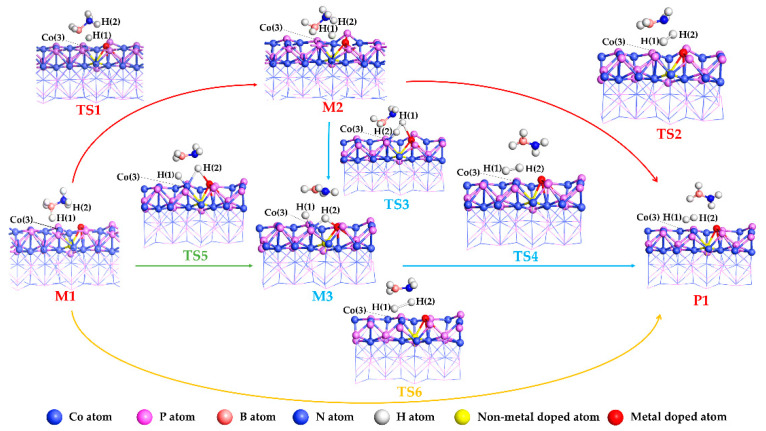
Mechanism of hydrogen evolution reaction of NH_3_BH_3_.

**Figure 4 molecules-27-08206-f004:**
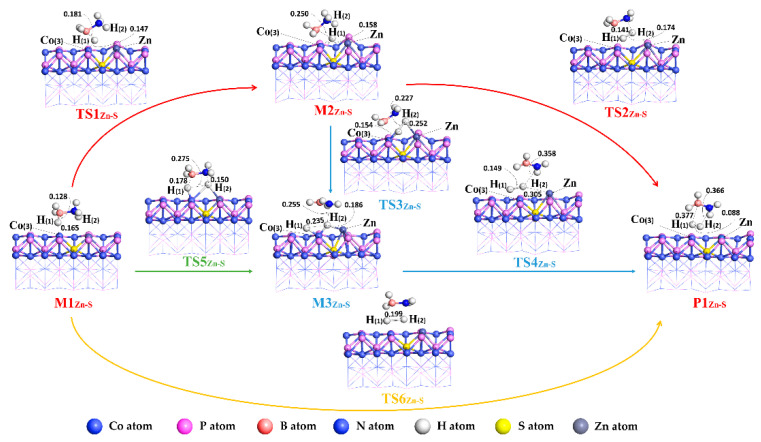
General flowchart of NH_3_BH_3_ hydrogen production reaction on CoP_Zn-N_ (101) surface.

**Figure 5 molecules-27-08206-f005:**
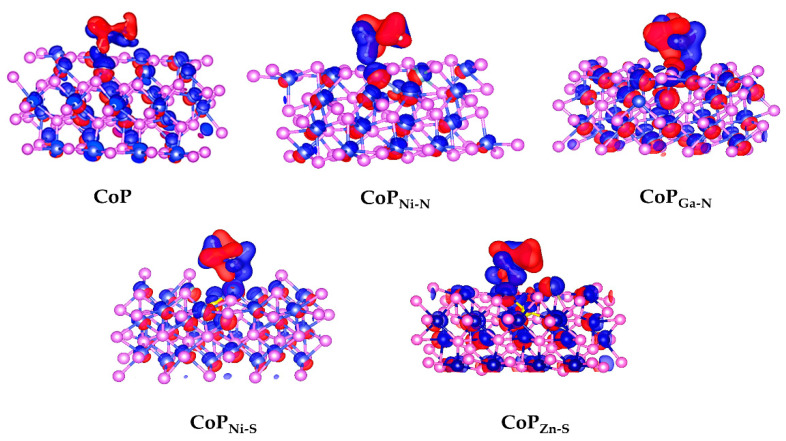
CDDs maps of CoP, CoP_Ni-N_, CoP_Ga-N_, CoP_Ni-S,_ and CoP_Zn-S_ catalyzed NH_3_BH_3_.

**Table 1 molecules-27-08206-t001:** The bond length (Å) parameters of reaction sites in CoP_Zn-S_-catalyzed NH_3_BH_3_ hydrogen evolution process. (Å).

Pathway		B-H(1)	Co(3)-H(1)	N-H(2)	Co(3)-H(2)	Zn-H(2)	H(1)-H(2)
I	M1_Zn-S_	1.28	1.65	1.03	--	--	2.53
	TS1_Zn-S_	1.81	1.47	1.02	--	--	2.19
	M2_Zn-S_	2.50	1.58	1.05	--	--	2.41
	TS2_Zn-S_	2.61	1.52	1.92	--	--	1.72
	P1_Zn-S_	3.77	1.62	3.66	--	--	0.88
II	M1_Zn-S_	--	1.65	1.03	3.45	3.03	2.53
	TS1_Zn-S_	--	1.47	1.02	3.29	2.66	2.19
	M2_Zn-S_	--	1.58	1.05	3.31	2.96	2.41
	TS3_Zn-S_	--	1.54	1.51	3.15	2.52	2.27
	M3_Zn-S_	--	1.65	2.55	1.60	1.86	2.35
	TS4_Zn-S_	--	1.63	3.58	1.59	3.05	1.49
	P1_Zn-S_	--	1.62	3.66	1.61	3.86	0.09
III	M1_Zn-S_	1.28	1.65	1.03	3.45	3.03	2.53
	TS5_Zn-S_	1.78	1.62	1.50	2.78	2.37	2.75
	M3_Zn-S_	3.33	1.63	2.55	1.59	3.05	2.35
	TS4_Zn-S_	3.83	1.63	3.58	1.59	3.05	1.49
	P1_Zn-S_	3.77	1.62	3.66	1.61	3.86	0.88
IV	M1_Zn-S_	1.28	--	1.03	--	--	2.53
	TS6_Zn-S_	2.30	--	1.91	--	--	1.99
	P1_Zn-S_	3.77	--	3.66	--	--	0.88

**Table 2 molecules-27-08206-t002:** The each position energies (E), relative energies (E_rel_) and activation energies (E_a_) of ammoborane reaction catalyzed by CoP_Zn-S_.

Pathway	Compound	E_rel_	E_a_
		kcal/mol	kcal/mol
pathway I	M1_Zn-S_	0.00	
	TS1_Zn-S_	22.71	22.71
	M2_Zn-S_	−4.83	
	TS2_Zn-S_	37.21	39.56
	P1_Zn-S_	−29.93	
pathway II	M1_Zn-S_	0.00	
	TS1_Zn-S_	22.71	22.71
	M2_Zn-S_	−4.83	
	TS3_Zn-S_	37.21	42.04
	M3_Zn-S_	−29.93	
	TS4_Zn-S_	−20.20	9.73
	P1_Zn-S_	−26.62	
pathway III	M1_Zn-S_	0.00	
	TS5_Zn-S_	28.80	28.80
	M3_Zn-S_	−29.93	
	TS4_Zn-S_	−20.20	9.73
	P1_Zn-S_	−26.62	
pathway IV	M1_Zn-S_	0.00	
	TS6_Zn-S_	22.15	22.15
	P1_Zn-S_	−26.62	

**Table 3 molecules-27-08206-t003:** The reaction pathway activation energies of five catalysts in pathway I–IV (kcal/mol).

Pathway	Compound	CoP	CoP_Ni-N_	CoP_Ga-N_	CoP_Ni-S_	CoP_Zn-S_
pathway I	TS1	21.88	29.15	22.48	20.67	22.71
	TS2	51.65	44.87	65.63	45.16	39.56
pathway II	TS1	21.88	29.15	22.48	20.67	22.71
	TS3	31.35	35.81	23.18	19.14	42.04
	TS4	0.57	5.27	6.67	7.40	9.73
pathway III	TS5	36.68	33.45	47.29	47.94	28.80
	TS4	0.57	5.27	6.67	7.40	9.73
pathway IV	TS6	52.02	27.11	44.19	48.48	22.15

## Data Availability

The data presented in this study are available on reasonable request from the corresponding author.

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
