# Peer review of "Theoretical Study of Hydrogen Production from Ammonia Borane Catalyzed by Metal and Non-Metal Diatom-Doped Cobalt Phosphide"

_molecules, 2022, doi:10.3390/molecules27238206_

Round 1

Reviewer 1 Report

The authors designed four dual-doped CoP and studied the catalylic performance of these catalysts on the pyrolysis of NH3BH3 by DFT calculation. The highlight of the manuscript is the detailed discussion on the mechanism, which will be helpful for the readers to deeply understand the catalytic behaviors. I am willing to recommend the publication of the manuscript. However, some revisions should be made before the manuscript can be accepted for publication. Following are some suggestions. 

1. It is well-known that it is necessary to consider the value of U in the case of Co, whether the authors discussed the influence of U?

2. Does the authors selected the doped site random? In figure 1, the doped site is not unique. It is better to discuss which the most favorable conformation.

3. For the mechanism discussion, it is better to compare the results with available references. Moreover, the discussions of DOS, band structure and CDD should combined with the discussions of mechanism.

4. The RDS changed with the catalysts, it is better to explain it in view of the electronic structure of catalysts. Moreover, the pyrolysis of NH3BH3 carried out via mechanism IV in the case of CoPNi-N and CoPZn-S catalyzed reaction. Is there any reference supported the proposed mechanism IV?

5. Some recent papers on hydrogen production from NH3BH3 are suggested to be cited so that the readers can better understand the background of this work. Applied Catalysis B: Environmental 320 (2023) 121973; ACS Appl. Mater. Interfaces 14(2022) 2797927993.

Author Response

Reviewer #1:

The authors designed four dual-doped CoP and studied the catalylic performance of these catalysts on the pyrolysis of NH3BH3 by DFT calculation. The highlight of the manuscript is the detailed discussion on the mechanism, which will be helpful for the readers to deeply understand the catalytic behaviors. I am willing to recommend the publication of the manuscript. However, some revisions should be made before the manuscript can be accepted for publication. Following are some suggestions.

Author Reply: Thank the reviewer for such positive comments.

  1. It is well-known that it is necessary to consider the value of U in the case of Co, whether the authors discussed the influence of U?

Author Reply: Thank you for your valuable advice. We have considered the value of U and carried out tests on the energy of the adsorption model and the adsorption basis respectively. Through our test, it is found that whether U is added or not has little influence on the reaction energy difference, which is only 0.04 eV, which can be ignored. We also refer to the other theoretical research articles about cobalt phosphide catalysts. In these articles, authors also did not consider the value of U in calculation. As follows: ChemElectroChem 10.1002/celc.201800601; RSC Adv., 2021, 11, 2947; Nano Research 2022, 15(10): 8897−8907.

  1. Does the authors selected the doped site random? In figure 1, the doped site is not unique. It is better to discuss which the most favorable conformation.

Author Reply: Thank you very much, the selection of doped sites is based on previous related researches. Due to the symmetry and periodicity of the surface atoms, the choice of sites may have a slight influence on the reaction. However, the focus of this study is the influence of different types of doped on the reaction, which means in the case of the same doped sites, the influence of different types of diatom-doped on the catalytic effect was mainly considered, and the influence of diatom-doped site selection on the reaction is not considered for the moment. Thank you very much for your valuable advice. We will continue to discuss the differences caused by the different doped sites of each type in the subsequent work.

  1. For the mechanism discussion, it is better to compare the results with available references. Moreover, the discussions of DOS, band structure and CDD should combined with the discussions of mechanism.
  2. Author Reply: Thanks for your kindly reminder. We have added some discussions in our paper “Many studies on cobalt phosphide modified materials can prove that the dopant of N, S, Ni, Zn and Ga can improve the catalytic performance of CoP, which are consistent with our theoretical calculation results. For instance, Chen et al.[53] reported that Ni-doped CoP could accelerate the process of hydrogen evolution both in acid and alkailne mediums, showing excellent electrochemical stability and durability. Li et al. [54] found N and Mo co-doped heteroatoms can optimize the morphology and surface structure of CoP. Anjum et al.[55] studied sulfur-doped cobalt phosphide electrocatalyst and concluded that its performance is better than all-noble-metal electrocatalysts in alkaline electrolyzer for overall water splitting.Yang et al.[56] synthesized Zn doped CoP nanowire arrays for boosting hydrogen generation, and they find the overpotential of Zn doped CoP was two times lower than undoped CoP. Zhang et al.[57] also reported that Ga dopant could enhance the activity of CoP“.in the second paragraph on page 7.“Sun et al. also reported similar studies on polysulfur confinement and electrochemical kinetics of amorphous cobalt phosphide-enhanced lithium-sulfur batteries[57]. in the first paragraph on page 9.

  1. The RDS changed with the catalysts, it is better to explain it in view of the electronic structure of catalysts. Moreover, the pyrolysis of NH3BH3 carried out via mechanism IV in the case of CoPNi-N and CoPZn-S catalyzed reaction. Is there any reference supported the proposed mechanism IV?

Author Reply: Thanks, In this paper, by calculating the activation energies of the control steps of the four reaction pathways, the optimal reaction pathways of each catalyst were found. So far, we have only calculated the RDS of different catalysts at a shallow level and found the optimal reaction path through comparison. Further discussion may need to be conducted in follow-up studies. Electronic structure of catalysts, such as the band structures and density of states of the five catalysts were obtained. The proposed mechanism IV comes from previous studies. You can refer International Journal of Hydrogen Energy 38, Issue 1, 11 January 2013, Pages 169-179; Surface Science 680, February 2019, Pages 95-106.

  1. Some recent papers on hydrogen production from NH3BH3 are suggested to be cited so that the readers can better understand the background of this work. Applied Catalysis B: Environmental 320 (2023) 121973; ACS Appl. Mater. Interfaces 14(2022) 27979−27993.

Author Reply: Thanks for your kindly reminder. We have read the two cutting-edge literature you sincerely recommended and then cited them in our paper. Meanwhile, we also cited some classic literature as follows, so that readers can better understand our research background. ACS Appl. Nano Mater. 2021, 4, 8, 7640–7649; ACS Appl. Energy Mater. 2021, 4, 1, 633–642; Chemical Engineering Journal 449, 1 December 2022, 137755; We have added some discussions in our paper. At present, quantities of composite materials with high catalytic activity and high stability which can catalyze the evolution of hydrogen from ammonia borane have been synthesized[20-24]. in the first paragraph on page 2.

Reviewer 2 Report

The paper presents the results of the theoretical study of the decomposition of ammonia borane on a series of CoP-based solid catalysts, which is relevant in the context of transitioning to a cleaner energy production. The authors used density functional theory which is a state of the art methodology to study chemical reactions, in conjunction with transition state theory.

Overall, the formulation of the objectives, methodology and results are clearly presented, and the conclusions are in line with the results obtained. 

Author Response

Thanks a lot for your such positive comments.